# QoS-Aware Scheduling Algorithm Enabling Video Services in LTE Networks

**Amal Abulgasim Masli [1], Falah Y. H. Ahmed [2,\*] and Ali Mohamed Mansoor [3]**

[1] Faculty of Information Science and Engineering, Management and Science University, Shah Alam 40100, Malaysia; amalabulgasim@gmail.com
[2] Faculty of Computing and Information Technology, Sohar University, Sohar 311, Oman
[3] Department of Software Engineering, Faculty of Computer Science & Information Technology, University of Malaya, Kuala Lumpur 50603, Malaysia; ali.mansoor@um.edu.my
\* Correspondence: fhamod@su.edu.om

**Abstract:** The Long-Term Evolution (LTE) system was a result of the 3rd-Generation Partnership Project (3GPP) to assure Quality-of-Service (QoS) performance pertaining to non-real-time and real-time services. An effective design with regards to resource allocation scheduling involves core challenges to realising a satisfactory service in an LTE system, particularly with the growing demand for network applications. The continuous rise in terms of the number of network users has resulted in impacts on the performance of networks, which also creates resource allocation issues when performing downlink scheduling in an LTE network. This research paper puts forward a review of optimisation pertaining packet scheduling performance through the LTE downlink system by introducing a new downlink-scheduling algorithm for serving video application through LTE culler networks, and also accounts for QoS needs and channel conditions. A comparison of the recommended algorithms' performances was made with regards to delay, throughput, PLR, and fairness by utilising the LTE-SIM simulator for video flow. On the basis of the outcomes obtained, the algorithms recommended in this research work considerably enhance the efficacy of video streaming compared against familiar LTE algorithms.

**Keywords:** LTE network; downlink scheduling; video streaming; QoS; EXPRule; jitter

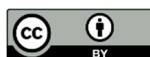

## 1. Introduction

Long Term Evolution (LTE) is recommended as a mobile technology since it counters the growing demand for diverse data streaming services and backs an extensive gamut of multimedia and online services, such as Internet TV, file sharing, web browsing, VoIP, wireless sensors, and video streaming, even in high mobility scenarios [1]. LTE has been designed considering downlink transfer technology based on Orthogonal Frequency Division Multiple Access (OFDMA) [2] and by employing different Radio Resource Management (RRM) procedures, with the objective of achieving large bandwidths and higher-level modulation (up to 64QAM), as well as spatial multiplexing, which can be obtained when there is a high downlink data rate in order to support higher spectral efficiency, reductions in packet delay and Packet Loss Ratio (PLR), and high data rates, compared with previous 3G networks [1,3]; moreover, it is plausible that it is swiftly moving towards 5G wireless technology [4,5].

The users of networks judiciously grade the quality-of-service in terms of throughput, delay, packet loss, and jitter, particularly for time-sensitive web-based services. Due to this, high standards have been implemented for QoS needs in order to enable real-time services that need real-time data [6]. Furthermore, designers and mobile operators face issues when offering QoS sustainably to support different services and applications with a range of QoS requirements [7]. Different applications have varying QoS requirements;

however, this research study concentrates on various parameters, including jitter, delay, throughput, and packet loss ratio, as well as making sure that different traffic flows use these requirements, which is also regarded as a key task, taking into account the continuing increase in the number of the network subscribers. Therefore, the delay factor is considered crucial for various multimedia applications, such as video conferences, IPTV, VoIP, and VoLTE. Maintaining this at a specific level has an impact on the transmission delay-jitter in the network, while it also has a considerable effect on the QoS during real-time transmitting. The jitter is regarded to be a key parameter pertaining to QoS on wireless networks that also impacts data flow, which could also result in data packet loss when there is buffer overflow, due to delays in processing the data; this ultimately results in the deterioration of QoS [7]. Thus, to address the issue of various network flows, it is crucial to study the impact of jitter and examine its behaviour. The researchers in [8] pointed out that real-time video transmission through a cellular network still encounters different challenges such as time-varying channel condition and inadequate bandwidth. Meanwhile, web-based video broadcasting is one of the most challenging services in network usage today. In earlier studies, researchers have focused on throughput, delay, and PLR, and used these as the key metrics to design the cellular networks. This research paper accounts for jitter in the LTE network, where a jitter can affect streaming video quality in the case of real-time services, as well as behaviour pertaining to video traffic in popular LTE downlink algorithms. To assess the performance of LTE networks with regards to throughput, delay, packet loss ratio and fairness, Proportional Fairness (PF), Exponential_ PF (PF), Exponential Rule (EXP-rule), and Modified Largest Weighted Delay First (MLWDF) were examined.

The rest of the paper is arranged as follows: the second section provides the state of art and the third section describes a downlink scheduling strategy in LTE networks. Section Four presents the implementation of scheduling rules by taking into account jitter value. The next section explains the simulations as well as the evaluation of performance. The conclusion is presented in the last section.

## 2. Related Work

In recent times, during the execution of interactive services in real-time, QoS requirements have grown and created congestion. Delay and jitter are regarded to be network performance indicators. Video quality correlates to network performance as well as the state of the received packets. Thus, just offering high network performance does not really ensure video quality. As cited in [9], a considerable impact is caused by the delay jitter on QoS versus packet loss and delay. The traditional scheduler algorithms do not account for the impact cast by network jitter on transmitting packets when it comes to real-time applications, such as online video conferencing or online gaming. The quality of viewing is significantly impacted by jitter. For such services, controlling beam jitter is regarded to be an indispensable part of maintaining QoS [10]. As per a recent paper [11] that examined the impact caused by jitter on video streaming in LTE networks, well-known downlink scheduling (PF, MLWDF, EXPRule) was clearly affected by jitter. QoS (QoS factors) can be defined as a group of different parameters that can impact the quality of video transmission. These parameters can easily impact the video quality of broadcasting and they also serve as measurement indicators for service providers. Indeed, even though the LTE radio network is regarded as decreasing latency compared to previous cellular technologies, it does not employ a fixed delay, as faulty transmissions are fixed via rapid re-transmissions [12]. In this case, transmissions occurring in the LTE network can result in jitter that can deteriorate the quality of applications in real-time. Thus, considering jitter as an aim and not as a limitation is key for certain applications, as mentioned in [13,14]. While delay jitter pertaining to data packets can impact the quality of experience for users, as stated in [15], it is also regarded as an essential QoS metric for real-time applications.

The difference pertaining to packet delay in a specific stream is measured via jitter, which is key for real-time interactive services such as video streaming and VoIP [16]. In

an ideal scenario, even though the packet delivery is set in a totally periodic manner and an equal reciprocal flow is generated by the source, the network produces an inevitable jitter because of the change in queue and propagation delay, and the arrival of the packets at the destination in different time range [17]. Thus, various researchers have focused on analysis and control of jitter in LTE networks in order to enhance performance. Mesbahi, and Dahmouni. [7] analysed behaviour pertaining to the jitter and delay in the LTE network by putting forward a jitter model that considered the service time for the total arrival rate for each data flow. The Poisson process was regarded as a traffic model and the results demonstrated that when the traffic load jitter and delay were increased, they behaved differently, wherein delay increased and the jitter reduced. Ref. [17] put forward a method to improve radio communications performance via radio level feedback. A jitter buffer was implemented to keep the levels of both delay-of-packet and PLR at a minimum. This method involved calculating a projected delay related to the radio events, receiving incoming transmission packets through a wireless communication medium, recognising radio events that would impact the timing of transmission packets to be received in the future, and holding and queuing the incoming transmission packets, as well as determining an effective delay as per the projected delay, wherein for controlling, an effective delay is employed on release of the queued incoming transmission packets. For heterogeneous downlink traffic LTE networks, [18] put forward a QoS-aware energy and jitter efficient model. In order to optimise the energy efficiency, C-RAN and RT-based scheduling models were employed, and packet delay jitter was used with a fixed delay budget that considered real-time behaviour pertaining to various types of traffic with services that demanded different quality requirements. Moreover, [15] investigated the energy efficiency EE pertaining to the UE, as well as the delay jitter over the LTE downlink. The study stressed the voice-over LTE traffic that is regarded as a thoughtfully-used service. For both EE and jitter delay due to the fixed delay budget, multi-target optimisation was carried out by utilising two low-complex exploratory algorithms. The results showed a basic comparison between the user equipment's energy efficiency versus the delay jitter packets. A recent work put forward in [18] focused on handling real-time High Definition (HD) video traffic with regards to delay jitter in LTE technology throughout the uplink channel. As per the related works, the delay jitter associated with video streaming is poorly assessed with regards to downlink scheduling in the LTE cellular network. Ref. [19] put forward enhancement for the EXPRule scheduler algorithm to improve the EXPRule algorithm by defining the drawback pertaining to this algorithm in terms of high delay and high packet loss ratio, as well as low fairness. This was achieved by integrating EXPRule, as well as an enhanced eEXPRule, in order to deliver considerable performance with regards to voice and video services in LTE downlink via decreasing head-of-line packet delay. This, in turn, is achieved through computation of transmission metrics with regards to traffic separately, which results in an overall increase in the throughput of UE.

## 3. LTE Schedulers

Providing satisfying QoS to active users is considered a key challenge for resource allocation with regards to every transmit time interval (TTI), since available resources need to be distributed on the network amongst users in order to cater to their requirements. LTE employs scheduling techniques in order to effectively use resources with regards to time and frequency bands [20]. For both uplink and downlink, packet scheduling mechanisms have been applied at eNBs for the MAC layer, in which allocation of the physical resources is done for downlink and uplink channels. Parts of the shared spectrum are assigned by the eNB to each user by complying with specific policies. The distribution of Resource Blocks (RBs) is controlled by the packet scheduler to cater to the needs of Users' Equipment (UE) and evade cell interference. In general, scheduling aims to achieve appropriate allocation pertaining to the basic physical resource block (e.g., frequency, time, power, etc.) to cater to users' equipment, which quenches the demand for QoS by users through specific scheduling patterns such as traffic type, channel condition, queue

status, and head-of-line packet delay, which is based on the prioritised packet scheduler decided by the users; it also determines which UE needs to be scheduled and assigned with regards to the PRBs [21]. Figure 1 illustrates the general model for downlink packet scheduling over LTE networks.

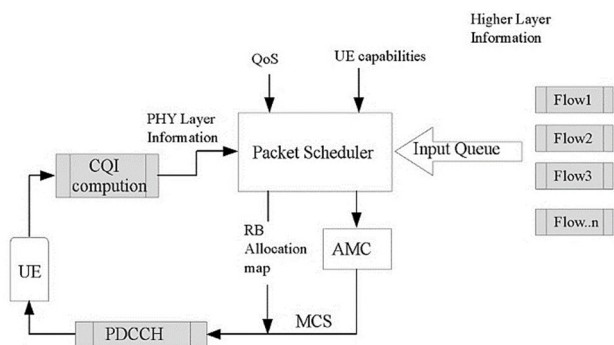

**Figure 1.** LTE DL packet scheduler general model.

Nonetheless, various issues exist with regards to the design of an LTE system solution and numerous downlink packet scheduling algorithms have been created to cater to QoS and fairness needs for different LTE optimisation services.

Thus, a brief explanation has been provided of the four different scheduling algorithms compared against the proposed method put forward in a simulation scenario, pertaining to MLWDF, PF, EXPRule, and EXP/PF.

### 3.1. Proportional Fair (PF) Scheduler

The PF algorithm is considered to be one of the best effort scheduling algorithms where QoS requirements are not guaranteed. The available radio resources are assigned via PF algorithm to the users, taking into account the experienced channel quality as well as the last value pertaining to user throughput. It can also be regarded as a weighting factor pertaining to the assumed data rate. The PF algorithm is designed to increase the total bit rate, as well as ensure fairness is maintained for the flows [3,22].

$$M_{i,k}^{PF} = \frac{d_{i,k}(t)}{R_i(t)}, \tag{1}$$

This metric also helps to determine the proportion that exists amongst the current accessible data rate $d_{i,k}(t)$ as well as the middling past data rate $R_i(t)$, with $i$ denoting the flow in $k$ flow sub-channel.

### 3.2. Modified Largest Weighted Delay First (MLWDF) Scheduler

This can be defined as a channel-conscious algorithm that supports numerous data users with a variety of QoS needs. MLWDF generally considers fairness and delay, as well as system throughput assurance. The handling of real-time and non-real-time flows is done differently. The PF with regards to real-time flow and non-real-time flow employing a weighing scale can be presented as follows:

$$M_{i,k}^{M-LDWF} = \propto_i \ D_{HOL,i} * M_{i,k}^{PF}, \tag{2}$$

$$\propto_i = - \frac{log \ log \ \delta_i}{\tau_i}$$

Here, $D_{HOL,i}$ signifies the head of line (HOL) packet delay pertaining to the user $i$ at a time t. The packet delay threshold is denoted as $\tau_i$, which takes into account each real-time flow. $\delta_i$ specifies the highest potential impact caused by the HOL packet delay on the user $i$ that would be greater than user's $i$ delay threshold. It should be noted that the

HOL packet is regarded to be the time contrast that would exist between the packet's arrived time and the present time [23].

### 3.3. Exponential PF (EXP/PF) Scheduler

The EXP/PF technique was designed to serve multimedia use cases with regards to systems employing time multiplexing [4]. The aim here is to improve real-time data flow priority when compared with non-real-time streams. At every scheduling interval, multiplication of the user number occurs due to various available sub-carrier sets for all network users to share concurrently [22]. For the overall delay pertaining to the sent packet, EXP/PF applies the exponential function as well as the PF characteristics. For best-effort data flow, EXP/PF is employed as PF, while processing of real-time data flow is done as EXP/PF in the following manner:

$$M_{I,K}^{EXP/PF} = \exp\left(\frac{\alpha_i D_{HOL,i} - x}{1 + \sqrt{x}}\right) \cdot \frac{d_k^i(t)}{R^i(t-1)}, \tag{3}$$

$$x = \frac{1}{N_{rt}} \sum_{i=1}^{N_{rt}} \alpha_i D_{HOL,i}, \tag{4}$$

where $N_{rt}$ denotes the number of active real-time DL flows.

### 3.4. Exponential Rule (EXPRULE) Scheduler

The algorithm is designed to increase user channel allocations to the maximum possible in varied channel conditions by analysing the network state in general. The EXP rule stipulates that a single user/queue is to receive service in every scheduling instance. Channel information is utilised that is queued without previous information on traffic access or statistics [22]. Order of priority in the EXP rule schedule therefore may be expressed in the following:

$$M_{i,k}^{EXPrlue} = b_i \exp\left(\frac{a_i D_{HOL,i}}{c + \sqrt{(1/N_{rt}) \sum_j D_{HOL,j}}}\right) \cdot \Gamma_k^i \quad , \tag{5}$$

where $a_i, b_i,$ and $c$ denote optimal parameters as required by the model. $N_{rt}$ denotes the number of downlink flows at RT, $D_{HOL,i}$ denotes the peak of line latency for user i, and $\Gamma_k^i$ represents the efficiency of spectrum for the *i*-th user on the *k* sub-channel.
$\alpha i$ and $bi$ are expressed as follows when c equals 1:

$$a_i \in \left[\frac{5}{0.99\tau_i}, \frac{10}{0.99\tau_i}\right], \tag{6}$$

$$b_i = \frac{1}{E[\Gamma^i]}, \tag{7}$$

## 4. Proposed PrOEXPRule Scheduler Design

In this paper, we propose a novel resource scheduling algorithm for the LTE downlink that optimises the EXPRule scheduler more efficiently. EXPRule was selected due to its high-level effect on transmissions across LTE networks, which accounts for throughput optimisation based on [22]. The EXPRule algorithm employs queues and channel data, but lacks statistical data concerning traffic channel use. The approach utilises information on the channel that is queued without previous information on arrival or channel statistics for traffic. The approach proposed in this paper offers improvements in QoS for video traffic under heavy load conditions. User channel state is basically analysed with consideration of the recorded jitter/delay characteristics and the maximum permissible delay for all packet transmissions to the end user, wherein scheduling decisions are performed every 1ms TTI as required in 3GPP specifications. The method of our proposed approach is to apply the calculated jitter values with the head-of-line (HOL) delay and required

delay for all packets, so as to improve channel states for all users at each TTI for every transmitted packet, even where channel conditions are too degraded to satisfy QoS requirements.

### 4.1. Simulation Environment

To assure our objective, the novel proposed PrOEXPRule schema was validated in one cell in an interference environment populated with different user numbers using the LTE-Sim simulator [24]. This module comprises open-source simulation software that is designed to model dissimilar scheduling strategies for both uplink (UL) and downlink (DL). User demand settings are provisioned as multi-user/multi-cell environments that cover user mobility, radio resources optimisation, frequency usage reconfiguration, adaptive modulation coding (AMC), and other relevant industrial and scientific aspects [25].

User mobility was analysed for every user; a simulation was performed to understand the average packet loss ratio, fairness, latency, and data rate. Every LTE downlink scheduling technique was simulated 150 times, and the data were used for the outcome. The proposed algorithm is constructed in a frequency domain experiment compares real-time video stream simulation outcomes beginning from the eNodeB base station and terminating at the user equipment. The schedulers used for this experiment employed the EXP_PF, PF, MLWDF, and EXPRule algorithms, using LTE-Sim to compare and measure performance.

The objective of this trace-specific traffic use case is to employ the video trace to send data to the streaming application. In this case, it is simulated using four downlink sequences. Understanding of the Media Access Control (MAC) components Queue, QoS, Radio Bearer, and Packet is required when executing the newly developed LTE-Sim algorithms.

LTE-SIM enables network simulation based on scenarios set by users. As video streaming was involved in our model, a single cell with interference case was utilised in a single-cell setting wherein a single base station was situated at the centre among four clusters with a 1 km radius. User equipment (UE) ranged from [5–30] at a uniform mobility speed of 120 km/h, with user intervals of 5. A video stream (encoded at 242 kbps) was analysed for each UE, with every stream active in simulated time. The simulation parameters are shown in the Table 1:

**Table 1.** Simulation parameters.

| Parameters | Value |
| --- | --- |
| Number of Simulators | 5 |
| Number of Clusters | 1 |
| Number of Cells in a Cluster | 4 |
| Initial User Count | 1 |
| User Interval | 5 |
| Maximum Users | 30 |
| Bandwidth | 5 MHz |
| RB Numbers | 25 |
| Slot Duration | 0.5 s |
| Scheduling Time (TTI duration) | 1 ms |
| Cell Radius | 1.5 km |
| Modulation Scheme | QPSK, 16QAM, 64QAM |
| Mobile Speed | 120 s |
| User Speed | 3 km/h |
| Video Bitrate | 256 kbps |
| Max Delay | 0.1 s |
| Flow Duration | 120 s |

| | |
|---|---|
| Simulation Duration | 120 s |
| Frame Structure | FDD |
| Video Bitrate | 242 kbps |

### 4.2. Proposed Optimisation Exponential Rule (PrOEXPRule) Scheduling

The algorithm proposed in this study is intended to enhance video stream data using properties such as PLR, fairness, throughput, and delay by addressing jitter/delay characteristics and considering the maximum permissible delay for every packet. The eNB base-station schedule provisions network resources depending on the user count present at the scheduling block (SB), which represents a single 1 ms TTI. An SB comprises two resource blocks (RB), which, in turn, comprise a subcarrier sequence with several OFDM symbols. For each TTI, the eNodeB resource scheduler performs allocations of radio resources according to the demands of active users, all of whom are competing for scarce network resources according to certain criteria. If the eNodeB has sufficient resources to meet all user demand, no problems will arise in the allocation of resources. Conversely, if the eNodeB has less resources than that required to meet all user demand, the scheduler will have a primary role in allocating resources. In this study, the key role of the eNodeB downlink scheduling algorithm is to lessen packet loss in the user network queue by analysing the recorded jitter values for all transmitted packets, in addition to applying the delay priority technique, in which fundamental QCI properties are used for specifying the maximum permissible packet delay and packet loss. User prioritisation is performed on the basis of the nearer-to-deadline approach. It is feasible to maintain a packet loss ratio below the threshold by polling the HOL delay corresponding to every user. Packets proximal to the delay threshold are given higher priority.

Generally, and for efficient decision-making in resource allocation in the downlink, the scheduler typically compares the metric values of each and every UE to every corresponding RB, RB $k_{th}$, which is allocated to the UE contingent on comparisons between metrics for the $i_{th}$ user with the largest $M_{i,k}$ value, with use of the following expression [24].

$$M_{i,k} = max_i \, M_{i,k}, \tag{8}$$

In addition to defined simulation parameters in Table1, the proposed scheduling takes into account the channel state by considering the jitter value, and its metric is computed by considering the mean transmission rate for every flow in real time. In the case of one buffer, there are F packet flows that are stored in a queue with different parameters, which may be the packet length distributions and the arrival time. With the aim of computing the jitter for flow f and as per [7], all flows are considered to have arrival rate $\lambda_i$ with distributions in service time $\mu_i$.

The total arrival rate is:

$$\lambda = \sum_{i=1}^{m} \lambda_i, \tag{9}$$

For the flows F, jitter in a single cell can be considered as convergent to:

$$J_i \approx \frac{1}{\eta} \left[ 1 - e^{\frac{-\eta}{\lambda_i}} \left( \frac{\eta}{\lambda_i} + e^{\frac{-\eta}{\lambda_i}} \right) \right], \tag{10}$$

where $= \lambda - \mu$.

The estimation of average transmission throughput Rf corresponding to the F flow and real-time throughput for user equipment corresponding to the k TTI subchannel is expressed as:

$$Rf\,(t) = 0.8\,Rf\,(t-1) + 0.2\,Rf(t), \tag{11}$$

$$\eta = Rf\,(t)\,\text{-}t \qquad, \tag{12}$$

$$e = \exp((-\eta)/\text{Rf}(t)), \tag{13}$$

$$J = \frac{1}{\eta} * [1 - e * (\frac{\eta}{R(t)} + e)], \tag{14}$$

For every scheduling interval, the user-specific HOL packet delay distance $(dp_k(t))$ is calculated; basic QCI characteristics are used to determine the maximum allowable packet delay and loss. Packet delay data is useful, as it can be used to meet delay and loss requirements (delay limit and HOL delay) along with CQI; user prioritisation is performed on the basis of the nearer-to-deadline approach.

$$dp_k(t) = T_k - D_k(t) \quad k\forall K, \tag{15}$$

where $T_k$ and $D_k(t)$ denote the delay limit and HOL packet delay for the k user corresponding to time $t$.

Scheduling is performed to serve the user with the least $dp_k(t)$:

$$W = \arg\min dp_k(t) \quad k\forall K, \tag{16}$$

The best RB value (largest real-time downlink CQI metric provided by the user) $q_{i,max}(CQI_{k,n})$ is identified from the RB set. User data is transmitted on the RB and the $dp_k(t)$ value is updated. Subsequently, the chosen RB is eliminated from the available RB list.

The k user corresponding to the nth RB has a resource allocation identified by $a_{k,F}$.

The feasible data rate $r_k$ corresponding to the k user's subframe is expressed as:

$$r_k = \sum_{k=1}^{K} a_{k,F} \sum_{k=1}^{k} J_k(t). \sum_{k=1}^{q_{i,max}(CQI_{k,n})} D_k(t)., \tag{17}$$

under the following assumptions:

- An RB will be provided to a user if $a_{k,F} = 1$ and $a_{k',F} = 0 \ \forall \ k' \neq k$;
- $D_k(t)$ denotes the HOL packet delay corresponding to time $t$ and user $k$, and $D_k(t) > dp_k(t) \ \forall \ k$;
- $J_k(t)$ denotes the packet jitter corresponding to the transmission at time t and user k.

As the proposed PrOEXPRule technique aims to take into account the QoS factor in the functioning of video services, the metric for transmission is expressed as:

$$M_{i,k}^{ProEXPRule} = max_{a_{k,f},c_{k,c}} \sum_{k=1}^{k} \sum_{f=1}^{F} a_{k,F} \left[ \left( M_{i,k}^{EXPrule} / \sum_{k=1}^{k} J_k(t) \right). \sum_{k=1}^{q_{i,max}(CQI,n)} D_k(t). \right]. \tag{18}$$

This expression defines the objective function for the overall achievable data rate for the current TTI.

Algorithm PrOEXPRule (Algorithm 1)

| **Algorithm 1:** The proposed PrOEXPRule Scheduling. |
| --- |
| 1 input |
| 2 k //user equipment count; |
| 3 N$_{rb}$ //number of resource block RBs; |
| 4 r$_k$ //data rate required for user k; |
| 5 P$_k$ //priority index n for user k (0 or 1); |
| 6 ar //RBs available for serving each user; |
| 7 N$_k$ //estimated number of RBs required by each user; |
| 8 D$_k$ //head of line delay per user k; |
| 9 T$_k$ //delay threshold per user k; |
| 10 $\lambda_j$ //arrival data rate for each user; |
| 11 $\mu_j$ //service time; |
| 12 r$_k$ //achievable data rate; |

13 ARp//RBs allocated to non-priority users;

14 ARnp//average channel gain;

15 Jj   //jitter for each packet unit j;

16 initialisations:

17 define L for RBs list at each TTI;

18 define F for selected flow list that will be scheduling at each TTI;

19 $\eta = \lambda - \mu$

20 $S_K = \{\ \ \}, K \in \{1,2,3,\ldots\ldots,K\};$

21 $a_{k,f} = 0, k \in \{1,2,3,\ldots.,k\} \& f \in \{1,2,3,\ldots F\};$

22 $r_k = 0, \quad k \in \{1,2,3,\ldots,k\};$

23 $calculate \ \ avg = \frac{\sum_{k=1}^{B_{cqi}} avg_k}{N_{rb}}$

24 $calculate \ d_t = T_k - D_k$

25 $calculate \ N_k = N_{rb} \times \left(\frac{r_k}{avg_k}\right)$

26    sequence users based on their data rate requirements and $d_t$

27    M[i][j] = 0, max M[i][j] = 0;

28    $\overline{D_{HoL}} = D_{Hol,i};$

29    $\overline{R_t}(t+1) = (1-a).\bar{R}(t) + ar_i(t);$

30    TTI = 0;

31    while (TTI)

32           for j = 1 to R

33           for i = 1 to k

34         *if* $P_i$ = 1 then AR$_P$ hold; r$_k$

35           calculate: $J_i \approx \frac{1}{\eta}\left[1 - e^{\frac{-\eta}{\lambda_i}}\left(\frac{\eta}{\lambda_i} + e^{\frac{-\eta}{\lambda_i}}\right)\right];$

36             update: $\overline{D_{HoL}} = \frac{1}{N}\sum_{i=1}^{N} a_i D_{HoL,i};$

37             calculate: $dp_k(t) = T_k - D_k(t) \quad k\forall K$

38             update: $\bar{R};$

39           *select user with* min $d_t$ *and* max $r_k$

40           if ( $i \in infinite\_Buffer \ Flow$)

41         *if* $N_k \leq AR \ then$

42                   compute M[i][j] and allocate N$_k$ based on (17);

43           else

44                   compute M[i][j] based on $M_{i,k}^{ProEXPRule}$ (18);

45             end;

46       end;

47    end;

48 end;

## 5. Result and Discussion

In this part, the performance of PrOEXPRule, our proposed algorithm, is measured and assessed for video traffic. It is evaluated with respect to QoS parameters including system delay, throughput, fairness, and packet loss ratio. In order to assess our proposed LTE DL technique, it was compared to prevalent LTE scheduling techniques, i.e., EXP/PF, PF, M-LWDF, and EXPRule. The simulation scenarios were tailored based on a single-cell strategy and particular parameters as explained in Section 3.1. The results of the simulation of the proposed LTE DL technique PrOEXPRule are shown in Figures 2–5; these results illustrate the exact values that were gained from the simulator results, and these outcomes show that the proposed technique attained better performance with respect to QoS requirements and fulfilled the needs of video traffic streaming for the users of the network.

### 5.1. Video Packet Loss Ratio

Figure 2 presents the packet loss ratio PLR for video traffic for five different proposed algorithms. It clearly illustrates that the proposed PrOEXPrule algorithm achieved lower PLR than popular LTE DL algorithms. This is possible because packets reach their destination with an acceptable delay, maintaining the Quality of Service (QoS). Since PrOEXPrule achieved 0.00801 s with five users and EXP-Rule achieved 0.06268 s, the improved result of PrOEXPrule is clear. There was the same improvement with an increased number of users; at 25 users, the result for PrOEXPrule reached 0.19375, while EXPRule reached 0.61174 with the same number of users. Nevertheless, the PLR varied directly with the connected user count; the proposed technique offers superior results compared to the other techniques. MLWDF, EXP/PF, and EXPRule demonstrate better results than PF because they consider resource allocation delay before decision-making.

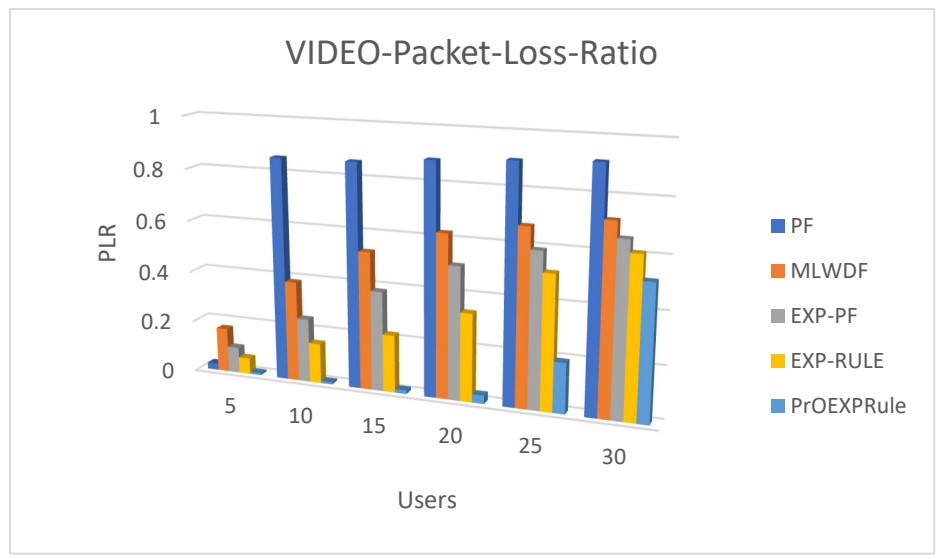

**Figure 2.** Packet loss ratio for video flow.

### 5.2. Video Fairness

Figure 3 depicts the fairness performance of the proposed technique. It is clear that the PrOEXPRule approach is superior to other techniques in terms of fairness because it provides better fairness and fixed throughput for all users, despite their increasing number. At 30 users, the EXPRule reached 0.31782, while the proposed PrOEXPrule achieved 0.32795 in throughput fairness for video flow; nevertheless, at 5 users all algorithms obtained close results between 0.39991 and 0.39989. The difference in results increase as the number of users increased, ranging from 0.13831 for the PF algorithm to 0.32795 for PrOEXPrule for 30 users. The PF algorithm has a low fairness index because throughput is compromised for fairness and vice versa. MLWDF, EXP/PF, and EXPRule presented a higher fairness index than PF. Hence, these algorithms have a satisfactory fairness index for video services at a specified level.

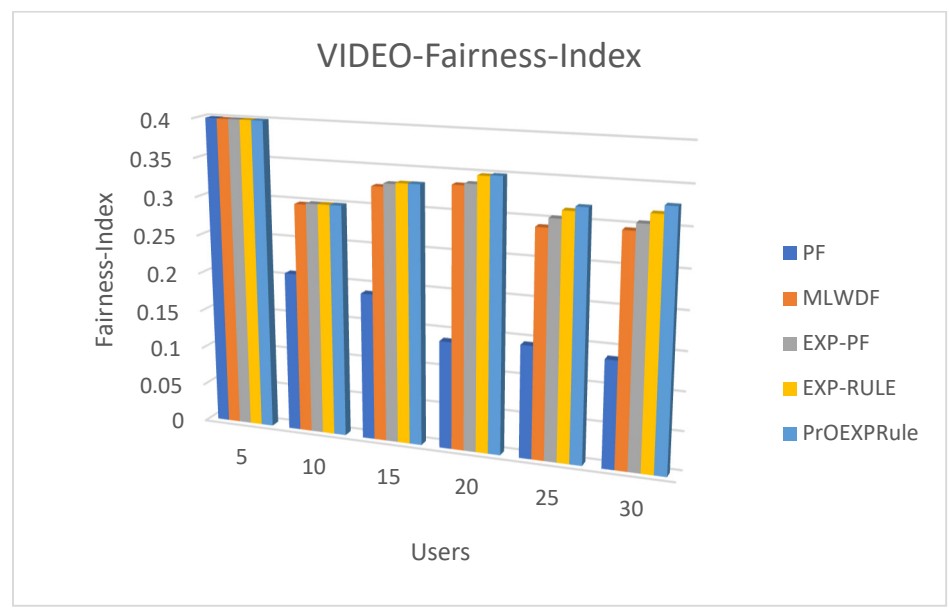

**Figure 3.** Packet fairness of video flow.

### 5.3. Video Delay

Figure 4 depicts video stream packet delay outcomes on the LTE network with increasing user load. The proposed technique provides less delay than other techniques because resource packet provisioning for video applications has the least PLR, as depicted in Figure 2. EXPRule, EXP-PF, MLWDF provide lower delay than PF. The PF technique supports non-real-time flow by providing fairness among users. When many resources blocks are provided video service traffic, the EXP-PF and EXP-Rule algorithms have relatively shorter delays than MLWDF, with 0.01723 and 0.01566 achieved for EXP-PF and EXP-Rule, respectively, at 20 users,. PrOEXPRule achieved 0.00415 at 20 users for packet delay; PrOEXPRule relies on exponential expressions and takes into account the HOL delay and delay priority, so it provides better results for delay compared to other algorithms.

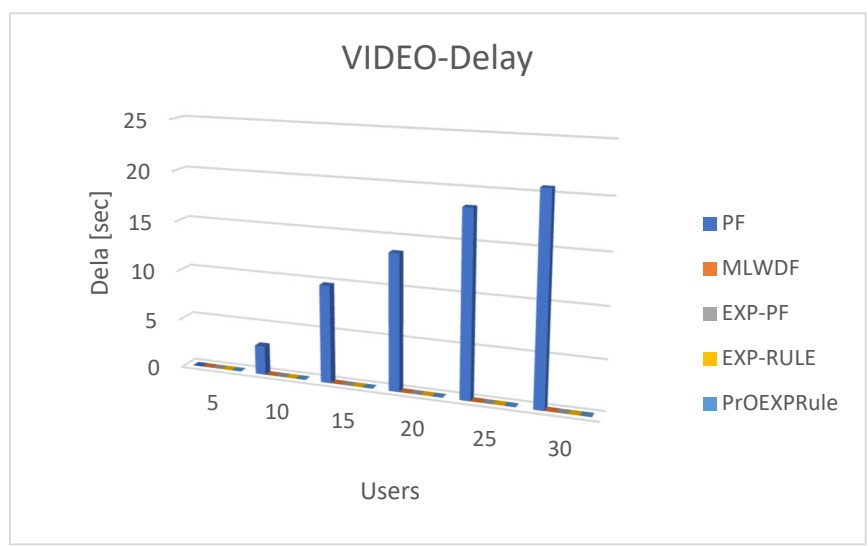

**Figure 4.** Packet Delay of Video Flow.

### 5.4. Video Throughput

Figure 5 presents the average data rate for video streams. The use of all algorithms is associated with a rise in average throughput as the user count increases. This is due to better resource distribution equity for users. The PrOEXPRule technique is superior to the

other techniques at optimising throughput. The algorithm proposed in this research has been verified to increase QoS fulfilment to satisfy user needs. When the user count is in the 5–20 range, the proposed algorithm reached 465,873.44000 bps, meaning that there is a sharp increase in average data rate that gradually decreases as the user count approaches the 20–25 range. When more than thirty users are connected to the network, the data rate using the proposed PrOEXPRule increased to 2,855,111.65333 bps. The corresponding figures for EXP-PF, PF, EXPRule, and MLWDF are 1,265,368.66667, 449,340.96000, 2,017,684.76000, and 1,707,329.85333 bps, respectively.

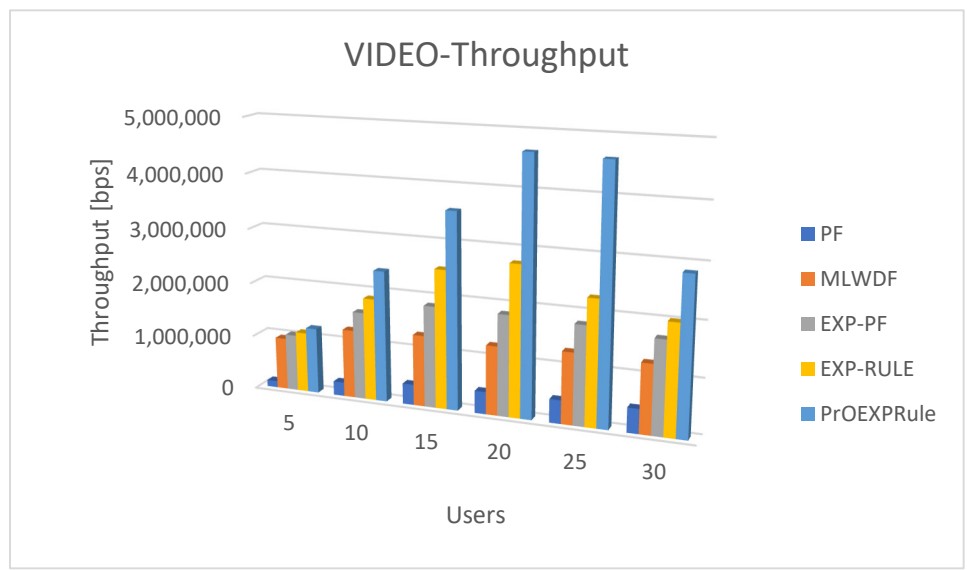

**Figure 5.** Packet throughput for video traffic.

## 6. Conclusions

A new improved scheduler that supports the LTE network downlinked direction is proposed in this study. It considers one of the crucial services over LTE networks, that is, video streaming, which is affected by different parameters including jitter, limited bandwidth, and delay, as considered in this study. To efficiently control the RBs allocation process and fulfil the QoS requirements, this study carried out a performance assessment of prevalent resource allocation techniques in LTE networks and proposes an improved scheduling technique focused on video streaming using the LTE-Sim simulator. The jitter index and delay threshold were regarded as the primary parameters to improve the video streaming service by taking the channel state and HoL delay into account. The proposed methods were validated by comparing the simulation results with well-known LTE DL scheduling algorithms, including PF, M-LWDF, EXP-PF, and EXP-rule schedulers. From the simulation outcomes, we can conclude that the proposed techniques performed better than the other algorithms. The efficiency of the proposed PrOEXPRULE technique improves video streaming performance with respect to fairness, delay, packet loss ratio, and packet throughput, even when there is an increased number of users. The results produced by the proposed algorithm demonstrated its ability to reduce delay and packet loss ratio, in addition to increasing packet fairness and throughput comparing to proposed LTE downlink algorithms.

**Author Contributions:** Formal analysis, A.M.M.; Supervision, F.Y.H.A.; Writing—review & editing, A.A.M. All authors have read and agreed to the published version of the manuscript.

**Funding:** This research received no external funding.

**Data Availability Statement:** https://telematics.poliba.it/index.php?option=com_content&view=article&id=28&Itemid=203&lang=en (accessed on 1 April 2022).

**Conflicts of Interest:** The authors declare no conflict of interest.

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
