# Peer review of "QoS-Aware Scheduling Algorithm Enabling Video Services in LTE Networks"

_computers, doi:10.3390/computers11050077_

Round 1

Reviewer 1 Report

  1. The proposed method description was not very clear.
  2. Some unites need to added to numbers in section 4.1, table 1
  3. In line 288, there is a word “chapter 3” , mostly it is typing mistake
  4. The graph style need to be changed in away that the data can be represent better.

Author Response

We are open to any further improvements in the paper. Please find the reviewer comments and the response.

Reviewer 2 Report

The article is interesting, however it needs some corrections. Below are some comments:

The authors have referred to selected algorithms in Section 3, which are compared with the algorithm proposed by the authors later in the paper. Unfortunately, the description of these algorithms is too laconic.

It is necessary to add a description of the EXPJRule algorithm in Section 4.3 for the reader to better understand its operation.

Do the authors have results of other simulations with different parameter values e.g. higher video bitrate. Were the same parameter values assumed for each UE, or were other varying values also tested?

The results presented in Section 5 need more explanation. For example, were the results obtained for each number of users exactly as shown in the graphs, or are these average values obtained during the simulation? What were the minimum and maximum values of the various parameters tested?

Correction and formatting of the manuscript is required.

Author Response

(The authors gave the same response as above.)

Reviewer 3 Report

The quality of presentation of the paper is so low such that the reviewer cannot read through it. Please check your submission format, font size and type for your mathematical equations, the quality of your demonstrated figures etc. such that the reviewers and readers can read your work easier. 

Author Response

(The authors gave the same response as above.)

Round 2

Reviewer 2 Report

I accept the provided explanations and introduced additions.

Author Response

We thank the journal and the editor for allowing us the opportunity to publish our work in Computers Journal and grateful for the helpful feedback by the Reviewers that helped us to improve the quality of the manuscript. We carefully responded to all points and have modified the manuscript accordingly. We are open to any further improvements in the paper. Please find the reviewer comments and the response.

Reviewer(s) Comments to Author: Recommendation: Major Revision

Reviewer 3 Report

The authors have not resolved properly the comments of the reviewer in the last round. The authors attached an under-edit manuscript with deletion marks etc. such that the reviewer cannot recognize if the quality of the revised manuscript had been improved.

Author Response

(The authors gave the same response as above.)
